# Eradication of avian leukosis virus subgroups J and K in broiler cross chickens by selection against infected birds using multilocus PCR

Alexander M. Borodin[1,2], Zhanna V. Emanuilova[1], Sergei V. Smolov[1], Olga A. Ogneva[1], Nina V. Konovalova[3], Elena V. Terentyeva[3], Natalia Y. Serova[4], D. N. Efimov[5], V. I. Fisinin[5], Anthony J. Greenberg[6]*, Yakov I. Alekseev[3,7]*

1 Breeding and Genetic Center Smena, Ministry of Science and Higher Education of the Russian Federation, Bereznyaki, Russia, 2 Institute of Medical and Biological Research, Nizhnii Novgorod, Russia, 3 Syntol LLC, Moscow, Russia, 4 All-Russian Research Veterinary Institute of Poultry Science Branch of the Federal Scientific Center All-Russian Research and Technological Poultry Institute Russian Academy of Science, St. Petersburg, Russia, 5 Federal Scientific Center All-Russian Research and Technological Poultry Institute Russian Academy of Science, Sergiev Posad, Russia, 6 Bayesic Research, Ithaca, NY, United States of America, 7 Institute for Analytical Instrumentation Russian Academy of Science, St. Petersburg, Russia

* tonyg@bayesicresearch.org (AJG); jalex@syntol.ru (YIA)

**Data Availability Statement:** All relevant data are within the manuscript.

## Abstract

The avian leukosis virus (ALV) is a serious threat to sustainable and economically viable commercial poultry management world-wide. Active infections can result in more than 20% flock loss, resulting in significant economic damage. ALV detection and elimination from flocks and breeding programs is complicated by high sequence variability and the presence of endogenous virus copies which show up as false positives in assays. Previously-developed approaches to virus detection are either too labor-intensive to implement on an industrial scale or suffer from high false negative or positive rates. We developed a novel multilocus multiplex quantitative real-time PCR system to detect viruses belonging to the J and K genetic subgroups that are particularly prevalent in our region. We used this system to eradicate ALV from our broiler breeding program comprising thousands of individuals. Our approach can be generalized to other ALV subgroups and other highly genetically diverse pathogens.

## Introduction

As the global population continues to grow, agricultural systems must increase production while maintaining sustainable levels of input consumption. To achieve this goal, we must control disease outbreaks in agricultural animal production facilities. Infections cause significant economic harm and contribute to global food insecurity. An important pathogen in poultry is the avian leukosis virus (ALV), a diploid single-stranded RNA virus that belongs to the genus *Alpharetrovirus* of the Retroviridae family [1]. ALV leads to lymphoid and myeloid leukosis, as well as neoplasia in other tissues. Active infections result in over 20% mortality, lower productivity, and cause significant harm to industrial poultry management [1]. Even subclinical

**Funding:** ZhVE, AMB, SVS: Ministry of Science and Higher Education of the Russian Federation, the state assignment for Breeding and Genetic Center Smena No. 075-01297-20-00. https://minobrnauki. gov.ru/ The funders had no role in study design, data collection and analysis, decision to publish, or preparation of the manuscript.

**Competing interests:** The authors have declared that no competing interests exist.

manifestations of ALV lead to decreased productivity and serious revenue loss [2, 3]. Spread of ALV presents significant problems worldwide. Several national eradication programs exist, for example in China, Australia, and Iran.

ALV types are classified into subgroups based on the GP85 envelope protein structure [4, 5]. Viruses from subgroups A through E [2], J [4], and K [5] are specific to chicken. Subgroup F and G viruses are specific to pheasants and are thus not relevant to the present study [6]. With the exception of the endogenous E group they are highly pathogenic. Endogenous virus genomes integrate into host chromosomes in the germline and are transmitted vertically via Mendelian inheritance [2]. ALV can integrate into somatic cell DNA and be induced by stress, leading to reversion of viremia [7]. Viruses belonging to the A, B, and J subgroups are common, while the C and D subgroups are encountered only rarely [2]. The K subgroup has been discovered relatively recently [5, 8–11]. Some of its variants are significantly pathogenic [12–15]. Studying the K subgroup emergence may lead to novel insights into the modes of virus spread and virulence evolution. Eradication programs typically focus on the J subgroup viruses. Russian poultry industry is severely affected by ALV belonging to this subgroup, with 70% of 223 production sites, surveyed in 46 regions, testing positive for anti-subgroup J (and 90% for all subgroups) antibodies [16]. There is evidence that susceptibility to infection has a genetic component, with a major susceptibility locus linked to the *ev21* locus. *ev12* and *ev21* produce complete endogenous viruses and associated with significant reductions in antibody response to ALV. These genes also predispose individuals to ALV shedding [17].

Fundamentally, the virus eradication problem in a breeding program must be solved by preventing vertical transmission to subsequent generations. This has been the foundation of previous approaches [18]. However, the high horizontal transmission rates exhibited by subgroup J viruses necessitates a more intensive screening [19]. The current standard involves serological tests of cloacal and vaginal swabs at 20 weeks, followed by viremic and blood serum ELISA at 22 weeks, a serological test of the first two eggs at 23 weeks, meconium of new-born chicks at 26 weeks, and finally sero-tests of egg albumen at 40 weeks [20]. Previous studies suggest that raising poultry in small groups and identifying infected chicks before hatching prevents horizontal transmission of ALV subgroup A [21] in layers and ALV J in broilers [22]. However, there is evidence that vertical transmission can occur even when specific viral antigens are undetectable [22, 23]. Furthermore, broiler chickens appear to be particularly susceptible to ALV subgroup J horizontal transmission [2], complicating the eradication process.

ALV identification via propagation in CEFs or DF-1 cells is currently the golden standard. This method suffers from severe drawbacks: it takes seven to nine days and requires significant investment in specialized equipment and laboratory space. An ELISA assay against a group-specific ALV *p27* antigen is much more widely used. However, it also has several disadvantages, chiefly a high false positive rate due to endogenous virus *p27* expression [24] and low sensitivity [2, 18].

DNA-based detection is possible because the RNA genome of the virus is reverse-transcribed upon cell entry, with subsequent host genome integration [25]. However, identifying ALV positive birds is complicated by the background noise coming from endogenous viruses and is compounded by exogenous virus genome variability. Several of published PCR-based protocols suffer from endogenous virus driven false positives. Nevertheless, PCR-based approaches are 15 to 20% more sensitive than culture or ELISA-based protocols [26]. The major difficulty in implementing a PCR protocol is the high variability of the ALV subgroup J target sequence. For example, there is up to 5.1% divergence between *gp85* DNA sequences isolated from the same individual [27]. Amino-acid identity among isolates can be as low as 86.2% [28], implying and even higher DNA sequence diversity. We describe a PCR-based system for identification of ALV subgroups A, B, J, and K. We overcame the DNA diversity

problem by using multiple loci and probes. We demonstrate the effectiveness of our method by using it as a screening method to identify infected individuals and thus eradicate ALV from our Smena8 broiler cross. Our approach can be easily deployed in typical industrial settings.

## Materials and methods

### Ethical statement

All samples included in this study were obtained from birds with official records. Sample collection did not involve animal killing and was performed in accordance with national and European regulations. No ethical approval was necessary.

### Test system development

We used BLAST (http://www.ncbi.nlm.nih.gov/BLAST) and ClustalW (http://www.genome.jp/tools-bin/clustalw) on sequences obtained from GenBank to find regions of the ALV genome that are conserved within but divergent between subgroups. GenBank entries used as reference sequences for region selection are M37980, HM452341(ALV-A); AF052428, JF826241 (ALV-B), J02342 (ALV-C), D10652 (ALV-D), EF467236, AY013303, AY013304, KC610517 (ALV-E); Z46390, JF951728, JQ935966, HM776937, JX855935, JF932002, KX058878, DQ115805, KX034517, KU997685, HM235668, HM582657(ALV-J), KF746200, KP686143, and GD14LZ (ALV K). Primers and probes for real-time PCR (Table 1) were developed based on published sequences [10, 29, 30] and synthesized by Syntol (Moscow, Russia). We took care to pick primers that did not amplify known endogenous ALV sequences. In particular, EAV-HP elements contain parts of ALV-J envelope gene. We chose primers outside of the common region, thus eliminating EAV-HP amplification.

### DNA isolation and quantitative real-time PCR

We isolated DNA from feathers (one feather per sample) using the M-Sorb-OOM kit from Syntol (catalog # OOM-502). Calamus fragments 0.3 to 0.5 cm long or chick cloacal smears (all birds from the Smena8 cross) were placed in 1.5 mL tubes and treated with 0.4 mL of the lysis buffer at 70˚C for 15 minutes while stirring. We separated debris by a three-minute centrifugation at 13 000 rpm. We transferred the supernatant to a fresh tube and continued the isolation protocol according the manufacturer's instructions. We used 1.5 $\mu$L of the resulting DNA solution in each PCR reaction. While screening the breeding population, individuals were re-tested up to eight times, unless determined to be positive and eliminated from the flock.

RNA isolation and reverse-transcription were performed using kits from Syntol (M-Sorb-OOM for extraction and OT-1 for reverse transcription). We used 1 feather from each bird. We confirmed quantitative PCR specificity by amplicon sequencing using primers ALVKF, SEQA-KR, SEQJF, SEQJR, ALVAF, and SEQA-KR on the Nanophore 05 genetic analyzer (Institute for Analytical Instrumentation RAS, St. Petersburg, Russia). Subsequent generations were tested for ALV-J and -K by pooling feathers from 10 birds in one sample.

We ran real-time PCR reactions on an ANK-48 machine (Institute for Analytical Instrumentation RAS, St. Petersburg, Russia). We ran 50 cycles, with the following steps: denaturation step (10 seconds, 93˚C), annealing and elongation step (30 s, 55˚C). We used 10 $\mu$L of the quantitative real-time PCR reaction mix from a commercial kit (M-428, Syntol, Moscow, Russia). Primer concentration was 450 nM and we used 100 nM of each probe per reaction.

**Table 1. Primers and probes used in this study.**

| Name* | Sequence (5'–3') | Amplicon length, bp | Subgroup | Purpose |
|---|---|---|---|---|
| ALVAF | GCCACACGGTTCCTCCTTAGA | 114 | ALV A | Multiplex primer |
| ALVAR | CGCAGTACTCACTCCCCATGAA | 114 | ALV A | Multiplex primer |
| APL | (5R6G)TACGGTGG(dT-BHQ1)GACAGCGGATAG-P | | ALV A | Multiplex probe |
| ALBF1 | GGCCGAGGCCTCCCCGAAA | 77 | ALV B | Multiplex primer |
| ALVBR | GTCTCATTAATTTCCTTTGATTGA | 77 | ALV B | Multiplex primer |
| BPL1G | (Cy5)CCCATGTACC(dT-BHQ2)CCCGTGCCTTG-P | | ALV B | Multiplex probe |
| JFF1F | GCCCTGGGAAGGTGAGCAAGA | 139 | ALV J | Multiplex primer |
| JJR | GGAAATAATAACCACGCACACGA | 139 | ALV J | Multiplex primer |
| JNP | (ROX)TCCTCTCGA(dT-BHQ2)GGCAGCAAGGGTGTC-P | | ALV J | Multiplex probe |
| JJPLN | (ROX)CAGCA AGGGTG(dT-BHQ2)CTTCTCCG-P | | ALV J | Multiplex probe |
| ALVKF | CGGAGCATTGACAAGCTTTCAGA | 72 | ALV K | Multiplex primer |
| ALVKR | GTGATTGCGGCGGAGGAGGA | 72 | ALV K | Multiplex primer |
| KPL | (Cy5.5)CCACCTCGTGAG(dT-BHQ2)TGCGGCC-P | | ALV K | Multiplex probe |
| ALV-JNF [9] | TTGCAGGCATTTCTGACTGG | 214 | ALV J | Published primer |
| ALV-JNR [9] | ACACGTTTCCTGGTTGTTGC | 214 | ALV J | Published primer |
| JCP [9] | (FAM)CCTGGGAAGGTGAGCAAGAAGGA-BHQ1 | | ALV J | Published probe |
| H5 [10, 11] | GGATGAGGTGACTAAGAAAG | 545 | ALV J | Published primer |
| H7 [10, 11] | CGAACCAAAGGTAACACACG | 545 | ALV J | Published primer |
| Probe [10, 11] | (FAM)CTCTTTGCAGGCATTTCTGACTGGGC(TAMRA) | | ALV J | Published probe |
| SEQA-KR | CGCGATCCCCACAAATGAGGAAA | 443 | ALV A | Sequencing primer |
| SEQA-KR | CGCGATCCCCACAAATGAGGAAA | 253 | ALV B | Sequencing primer |
| SEQJF | CCCTGGGAAGGTGAGCAAGAA | 498 | ALV J | Sequencing primer |
| SEQJR | CCTTTATAGCACACCGAACCGAA | 498 | ALV J | Sequencing primer |
| SEQA-KR | CGCGATCCCCACAAATGAGGAAA | 466 | ALV K | Sequencing primer |
| JEF | CCTATTCAAGTTGCCTCTGTGGA | 72 | ALV J | LTR primer |
| JER | GCTTGCTCTATTTGGCCGTCAGA | 72 | ALV J | LTR primer |
| JEP | (Cy5)CCATCCGAGC(dT-BHQ2)GCCTCCAGTCC-P | | ALV J | LTR probe |

*References provided for previously published primers

### ELISA

We used the IDEXX ALV-J Ab Test system (IDEXX Laboratories, Inc., USA) and the Avian leukosis virus antigen test kit from Synbiotics (USA) to perform ELISA on blood plasma samples from randomly selected birds according to the manufacturers' instructions.

## Results and discussion

### Multiplex quantitative real-time PCR of common ALV subgroups

We designed our multiplex test system to identify all common ALV subgroups. We focused on the locus coding for the *gp85* coat protein to target amplification of viruses of the A, B, J, and K subgroups (Table 1). Fig 1 shows primer positions and typical real-time fluorescence curves. We used synthetic DNA fragments corresponding to each subgroup, as well as field samples, as positive controls during test system development (Table 1). We performed serial ten- and subsequently two-fold dilutions of the positive control samples to estimate assay sensitivity and employed a previously described procedure [30, 31] to estimate sensitivity from our serial dilution data. Our essays can detect 25 genome-equivalents of A and B subgroup viruses, and 10 genome-equivalents of J and K.

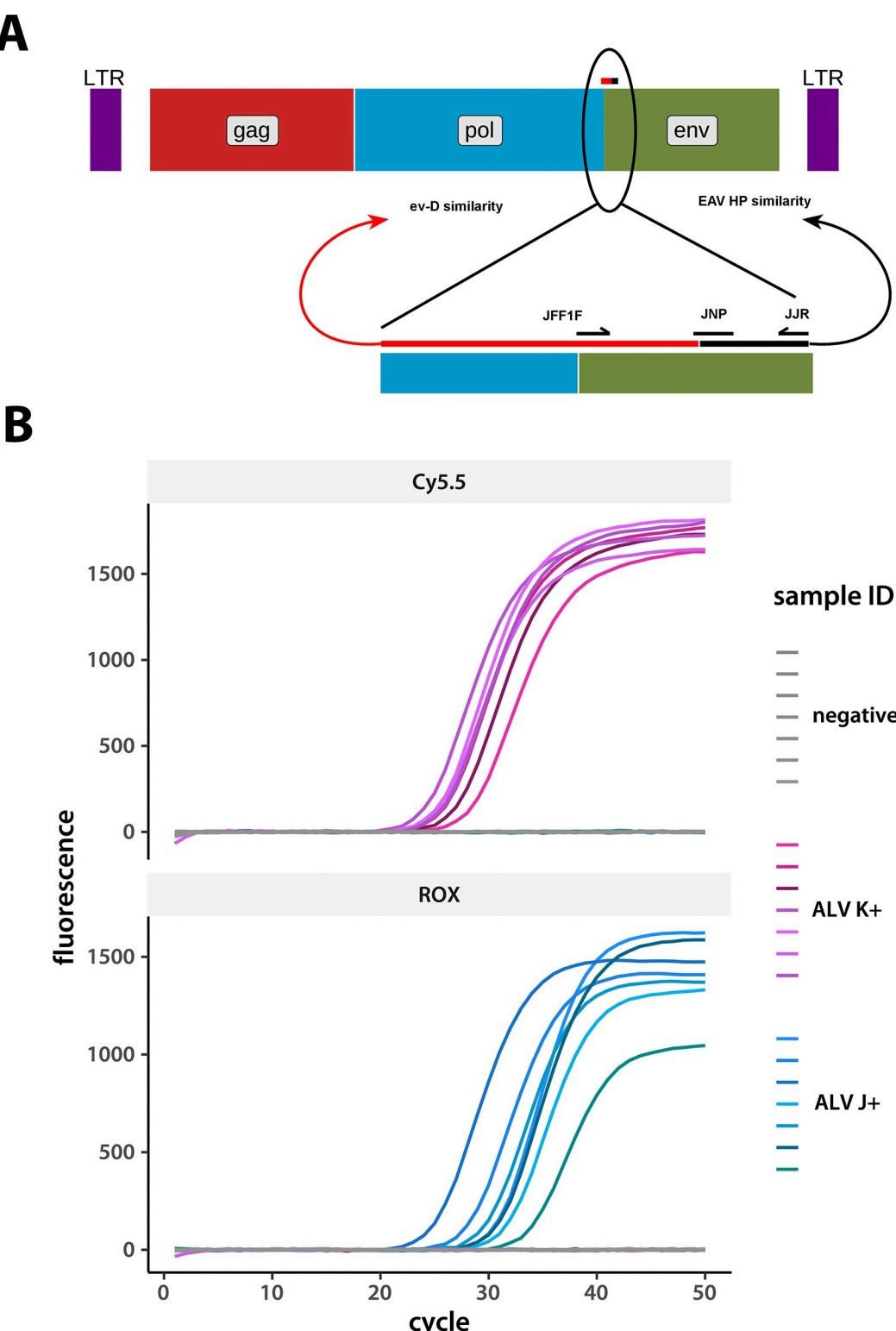

**Fig 1. Specificity of the developed real-time PCR system for detecting ALV J and K. A**. Primer positioning around the ev-D/EAV-HP (GenBank accession numbers DQ500016 and AC270426) similarity breakpoint in the ALV J gene that codes for the *gp85* coat protein. A diagram of the full ALV genome is on top, with the region surrounding primer placement shown in detail below. **B**. A representative set of detection curves from the Cy5.5 and ROX channels (the KPL and JJPLN probes, respectively, in Table 1). Each curve is a sample from a separate bird. Eight ALV J positive, ALV K positive, and

negative samples each were measured on the same 96-well plate and fluorescence values recorded simultaneously. Data for this graph are included as the S1 File.

To verify the performance of our system, we compared quantitative PCR threshold cycles obtained with the set of primers and probes specific to the J subgroup to previously published results (Table 1) [26, 29, 32]. We only used PCR methods for comparison since other approaches are unacceptably invasive for our application. Field isolates from poultry production facilities in the Moscow region were used for the latter estimates. We successfully verified our results using the Qin L system [26]. This system can detect fewer than 10 virus genome copies per reaction. We made the published primers ourselves (Table 1) and used them to replicate the previously described essays [26]. In our hands, the widely used primers H5 and H7 that detect the J subgroup genomes [29, 32] exhibit 100-fold lower sensitivity, consistent with previous reports [26]. Negative controls, either empty PCR buffer or specific pathogen free chicken DNA, gave no detectable signal after 50 cycles. We tested our identification system on a sample of 1200 individuals randomly picked from DNA isolates of four base lines used to initiate the Smena8 broiler cross. We found that up to 52% of all individuals in our flock were positive for the subgroup J ALV, while up to 10% carried subgroup K (some birds were double-infected). We did not detect any subgroup A or B cases. Previous studies [1] reported around 20% mortality in flocks affected by ALV. Given that not all infected birds die, our prevalence estimates appear to be in line with these previous observations.

We then moved on to analyze DNA from several tissues (feathers, liver, spleen, and blood) from the following regions in Russia: Moscow, Orenburg, Chelyabinsk, Kemerovo, Tyumen, Kaliningrad, Leningrad, Sverdlovsk, Novgorod, and Krasnodar. Samples collected in the Kaliningrad, Leningrad, Sverdlovsk, and Novgorod regions tested positive for K subgroup ALV. We found subgroup A infected individuals in the Leningrad region, while subgroup J positive samples were identified in areas surrounding Moscow, Sverdlovsk, and Leningrad. We did not detect any subgroup B positive samples, suggesting that this ALV variant was absent from Russia during the testing period. We confirmed our test system specificity by sequencing amplicons obtained from positive samples using primers described in Table 1. We did not see any subgroup misidentification.

## Multi-locus multiplex quantitative PCR and its use in ALV-J and K eradication

Having established a robust assay, we set out to test its effectiveness in eradicating the subgroup J and K infections we uncovered in our Smena8 broiler cross. To increase assay reliability, we implemented a multiplex approach, adding an extra probe specific to the *gp85* gene (JJPLN) and another that targets the long terminal repeat (LTR) sequence. Both probes are specific to the J subgroup viruses (Table 1). We succeeded in eradicating the virus infection by the 77th generation of the selection process.

We started by employing our J-subgroup specific PCR system for four generations of the Smena8 broiler cross (Table 1). High variability of the J genomes led us to implement the multi-locus multiplex system mentioned above, containing an additional probe (JJPLN, Table 1). This resulted in an average 2.5% increase in virus identification rates. We also introduced an additional amplification of a subgroup J-specific LTR fragment (Table 1), further increasing identification rates. The latter step allows us to see additional J subgroup variants. Managing the time interval between successive tests proved crucial. Increasing the between-test duration to 62 days between cycle four and five resulted in a two-fold jump in infection

rates. Therefore, from that point on testing was done no less frequently than once every two weeks. As a result, we no longer detected subgroup J viruses after cycle eight of the program in two Smena8 broiler cross lines (B7 and B9), while infection rates in two others were minimal (Table 2 and Fig 2). This in contrast to about 52% ALV subgroup J positive rate at the start of the program. Subgroup K viruses disappeared by cycle four, suggesting that they are easier to eradicate. Chickens from the next generation of the Smena8 broiler cross showed several ELISA-positive tests at day 267 after hatching. However, none of these were PCR-positive. We re-tested individuals from the generation after that at day 360 and found no positives by either PCR or ELISA. We saw no clinical manifestations of leukosis in the three years since. A handful of suspected cases proved to be PCR-negative.

**Table 2. ALV eradication in the 77[th] generation of Smena8 broiler cross pure lines.**

| Pure line | Age (days) | Livestock | Number infected | ALV K/J ratio | % infected |
|---|---|---|---|---|---|
| Line B5 Cornish | 1 | 1886 | 208 | N/A* | 11.0 |
| | 42 | 1032 | 112 | 35/77 | 10.9 |
| | 140 | 804 | 31 | 1/30 | 3.9 |
| | 155 | 775 | 7 | 0/7 | 0.9 |
| | 217 | 757 | 23 | 0/23 | 3.0 |
| | 239 | 610 | 57 | 0/57 | 9.3 |
| | 250 | 548 | 8 | 0/8 | 1.5 |
| | 265 | 531 | 2 | 0/2 | 0.4 |
| Line B6 Cornish | 1 | 2358 | 118 | N/A* | 5.0 |
| | 42 | 1142 | 47 | 8/39 | 4.1 |
| | 140 | 1088 | 58 | 1/57 | 5.3 |
| | 155 | 939 | 15 | 0/15 | 1.6 |
| | 217 | 923 | 28 | 0/28 | 3.0 |
| | 239 | 772 | 79 | 0/79 | 10.2 |
| | 250 | 687 | 5 | 0/5 | 0.7 |
| | 265 | 682 | 6 | 0/6 | 1.0 |
| Line B7 Plymouth Rock | 1 | 2319 | 134 | N/A* | 5.8 |
| | 42 | 1216 | 109 | 66/43 | 9.0 |
| | 140 | 966 | 135 | 5/130 | 14.0 |
| | 155 | 860 | 25 | 0/25 | 2.9 |
| | 217 | 841 | 69 | 0/69 | 8.2 |
| | 239 | 684 | 97 | 0/97 | 14.2 |
| | 250 | 602 | 10 | 0/10 | 1.7 |
| | 265 | 585 | 0 | 0/0 | 0.0 |
| Line B9 Plymouth Rock | 1 | 2466 | 67 | N/A* | 2.7 |
| | 42 | 1438 | 94 | 29/65 | 6.5 |
| | 140 | 1367 | 39 | 4/35 | 2.9 |
| | 155 | 1118 | 5 | 0/5 | 0.4 |
| | 217 | 1073 | 58 | 0/58 | 5.4 |
| | 239 | 919 | 71 | 0/71 | 7.7 |
| | 250 | 845 | 7 | 0/7 | 0.8 |
| | 265 | 823 | 0 | 0/0 | 0.0 |

*N/A—data are not available

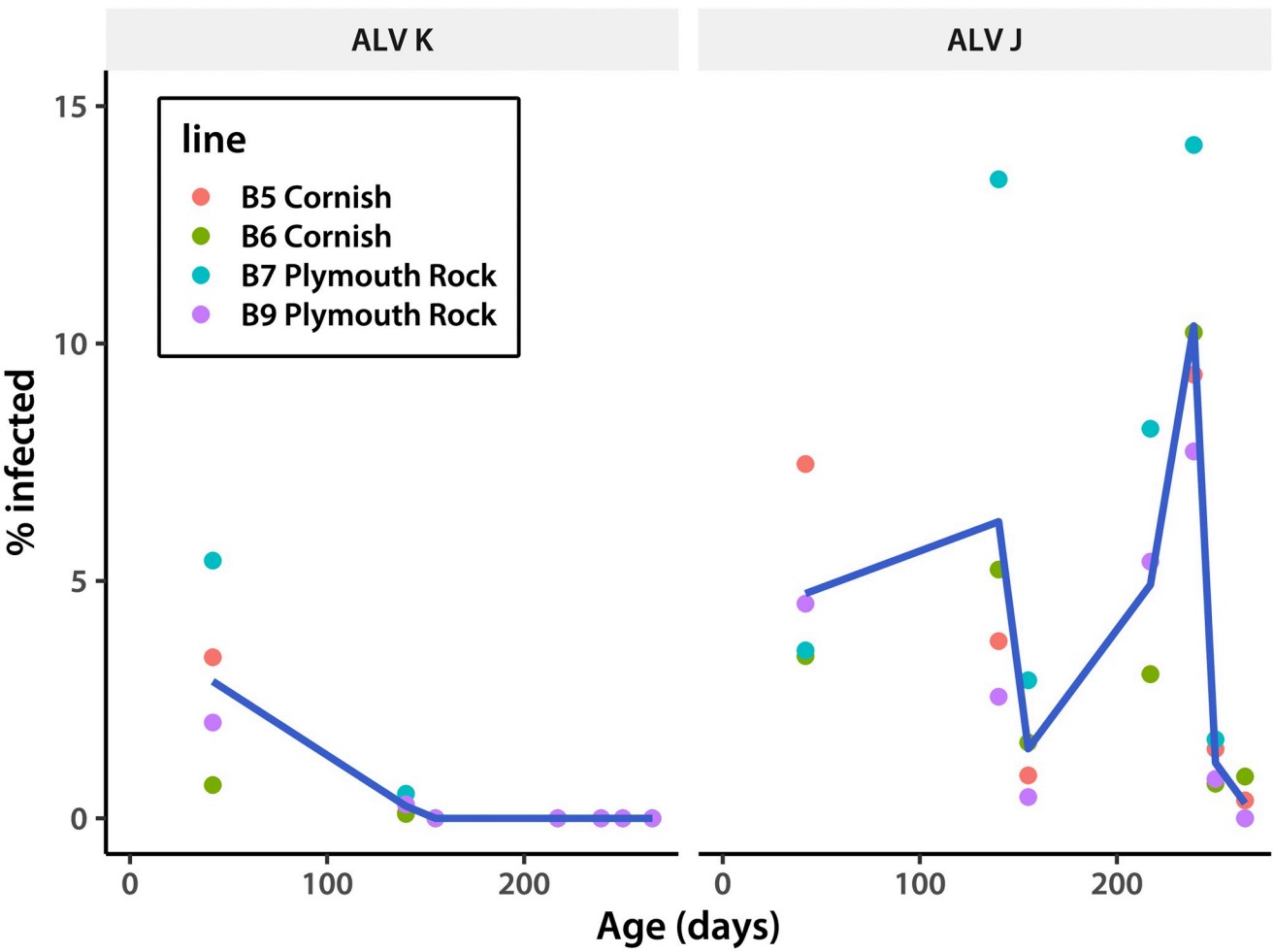

**Fig 2. Prevalence of infected individuals over time.** Data for ALV K and J infections are shown on separate panels. Lines trace among-line means, calculated for each day separately. The data for this plot are in Table 2.

## Conclusions

The avian leukosis virus is a serious threat to sustainable and economically viable commercial poultry management world-wide. As an RNA virus, it is highly variable, making reliable detection difficult. We developed a multi-locus multiplex quantitative real-time PCR system to identify individuals infected with subgroup J and K viruses with high sensitivity and specificity. We demonstrate the effectiveness of our approach by quickly eliminating both ALV J and K subgroups from our breeding program.

Our innovation is to couple quantitative PCR of multiple genes and the use of several probes per locus to increase sensitivity in the face of high genetic variation. In addition, using pulped feathers as the DNA source makes the assay non-invasive, cheap, and easy to implement [33–35]. Preliminary data indicate that we can get an almost two order of magnitude sensitivity increase when we assay ALV J cDNA, suggesting that future assay development can improve on the current approach, albeit perhaps at the cost of increased labor. Our experience in successfully eliminating ALV from our population comprising thousands of individuals demonstrates that the proposed system has adequate sensitivity and specificity. The procedure may be further simplified by screening only parents selected for establishment of the next

generation. This multilocus multiplex approach with multiple probes can also be used to detect retro- and coronaviruses, as well as any other highly variable microorganisms.

## Supporting information

**S1 File.**
(CSV)

## Acknowledgments

The authors are grateful to the two anonymous reviewers for comments that helped improve the manuscript.

## Author Contributions

**Conceptualization:** Alexander M. Borodin, Yakov I. Alekseev.

**Data curation:** Zhanna V. Emanuilova, Olga A. Ogneva, Nina V. Konovalova, Elena V. Terentyeva, Natalia Y. Serova, Anthony J. Greenberg.

**Formal analysis:** Alexander M. Borodin, Zhanna V. Emanuilova, Sergei V. Smolov, Olga A. Ogneva, Nina V. Konovalova, Elena V. Terentyeva, Natalia Y. Serova, D. N. Efimov, Yakov I. Alekseev.

**Funding acquisition:** Alexander M. Borodin, Zhanna V. Emanuilova, Sergei V. Smolov, Olga A. Ogneva.

**Methodology:** Alexander M. Borodin.

**Project administration:** Alexander M. Borodin, Zhanna V. Emanuilova, D. N. Efimov, V. I. Fisinin, Yakov I. Alekseev.

**Writing – original draft:** Alexander M. Borodin, Anthony J. Greenberg, Yakov I. Alekseev.

**Writing – review & editing:** Alexander M. Borodin, Anthony J. Greenberg, Yakov I. Alekseev.

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
