## [Decision Letter · Decision Letter 0]

15 Dec 2021

PONE-D-20-34696Complete eradication of avian leukosis virus subgroups J and K using multilocus qRTPCR in broiler cross chickensPLOS ONE

Dear Dr. Greenberg,

Thank you for submitting your manuscript to PLOS ONE. After careful consideration, we feel that it has merit but does not fully meet PLOS ONE’s publication criteria as it currently stands. Therefore, we invite you to submit a revised version of the manuscript that addresses the points raised during the review process.

I would encourage the authors to pay careful attention to the comments provided by the reviewers. Specifically, one of the reviewers identified that an identical paragraph has appeared in both the introduction and discussion which should be rectified. Both reviewers also raised concerns about how PCR could eradicate this virus, so I would suggest the authors consider changing the title and modifying this in the manuscript. 

We look forward to receiving your revised manuscript.

Kind regards,

Michelle Wille

Academic Editor

PLOS ONE

Journal Requirements:

2. Thank you for stating the following "The limited blood samples used in the study were taken during routine veterinary care. Infected animals were segregated from the general population but not culled." As part of your revision, we would ask that you include this within your Material and Methods section.

[This study was supported financially by Ministry of Science and Higher Education

 of the Russian Federation, the state assignment for Breeding and Genetic Center

 Smena No. 075-01297-20-00.]

 [ZhVE, AMB, SVS: Ministry of Science and Higher Education of the Russian Federation, the state assignment for Breeding and Genetic Center Smena No. 075-01297-20-00. https://minobrnauki.gov.ru/ The funders had no role in study design, data collection and analysis, decision to publish, or preparation of the manuscript.]

Additional Editor Comments:

I would encourage the authors to pay careful attention to the comments provided by the reviewers. Specifically, one of the reviewers identified that an identical paragraph has appeared in both the introduction and discussion which should be rectified. Both reviewers also raised concerns about how PCR could eradicate this virus, so I would suggest the authors consider changing the title and modifying this in the manuscript. 

Reviewers' comments:

Reviewer's Responses to Questions

**Comments to the Author**

1. Is the manuscript technically sound, and do the data support the conclusions?

Reviewer #1: Partly

Reviewer #2: No

2. Has the statistical analysis been performed appropriately and rigorously? 

Reviewer #1: N/A

Reviewer #2: I Don't Know

3. Have the authors made all data underlying the findings in their manuscript fully available?

Reviewer #1: No

Reviewer #2: No

4. Is the manuscript presented in an intelligible fashion and written in standard English?

Reviewer #1: Yes

Reviewer #2: Yes

5. Review Comments to the Author

Reviewer #1: Borodin and colleagues present new PCR assays for the detection of typical exogenous Avian Leukosis Viruses present in their flocks of interest. The authors clearly present the need for PCR-based assays to detect particularly subgroups J and K, but limited data is presented as evidence for the improvements made by their tests. Tables contain redundant information, and graphical representation of reductions in overall infection rate would be most welcome. The authors need to ensure they are consistent and clear in the differences between exogenous and endogenous ALV, and that the distinction between particularly J and K needs to be made clear when discussing results.

Of most particular concern is the repeat of an entire section of the introduction in the conclusion. This makes me doubt the integrity of the entire manuscript.

Lines 60-2 – Make clearer the distinction between highly pathogenic exogenous viruses and the much less pathogenic endogenous group.

Lines 74-5 – ALVE21/ev21 is not a gene, but an ALVE integration site, so please change this sentence. I think “ERV” will then be superfluous here, but it is the first time used (and therefore not defined).

Line 75 – the indicated reference 16 doesn’t talk about ALV susceptibility – is there an error here? Other works by Smith, Levin, Fadly etc from the early 1990s do talk about the importance of ALVE sequences (and not just ALVE21, but also ALVE6, ALVE9 etc) for receptor interference, but the relevance of receptor interference depends on the subgroup-specific viral entry receptors.

Lines 112-113 – I don’t yet understand how you can eradicate an exogenous virus by PCR alone? Unless eradicate just means to identify infected individuals, rather than eradicating the impact from the flock entirely?

Line 123,125 – GenBank, rather than GeneBank

Line 133 – The ALV-J envelope is derived from an EAV-HP element, often numerous in the genome. What precautions were taken to ensure other, non-ALV chicken ERVs were not detected?

Lines 170-2 – to put these types of figures on detection is interesting, but could it have context? Can you compare this directly to existing PCR and non-PCR approaches? OK, lines 177-179 do some comparison, but you say “in your hands” – can you give context of what has previously been reported in terms of sensitivity?

Lines 181-185 – 52% seems very high! Is this what you expected? Or higher than you suspected? Also (line 183) – switch herd for flock.

Line 186 – were there any differences in detection prevalence or apparent sensitivity between tissues?

Line 204-5 – Is the 77th generation what the authors mean? 77th round of PCR testing? No data is shown to support generational crosses?

Lines 235-247 (in conclusion) are a direct repeat of the introduction, only skipping the week number in each part of the third sentence. This is a very very odd thing to do – why have the authors done this? It makes me doubt other parts of the manuscript, if copy and pasting is being used…

Lines 248-9 – the authors don’t appear to show at any point any difference in detection between multiple site probes? So how can this assertion be made? It could be that one probe set detects all the time?

Tables1-4 – lots of the primers are repeated, creating superfluous content. Better to have one large table showing how different primers are used together and for what purpose, than many separate tables with duplicated information. A single landscape orientation page would cover it. Table legends could be more informative.

Table 5 give lots of information, but it’s really had to get the message across. A graph depicting the infection rate would be much more informative, or proportional ratios as a graph.

Reviewer #2: In the manuscript entitled “Complete eradication of avian leukosis virus subgroups J and K using multilocus qRTPCR in broiler cross chickens”, the authors developed a novel qRT-PCR assay to ALV subgroups A, B, J, and K . The authors worked hardly to validate their assays but

the following points should be considered and addressed by the authors in the manuscript prior to being submitted for publication as detailed below:

Major:

#Materials and Methods

-The M&M is very short and the authors should expand their methods to show for example: alignment figure where they highlight the locations of primers/probe.

-In DNA isolation and qRTPCR: how many feather samples were used? How many samples were used for RNA isolation? How many field samples? only 10?

-Description for how the authors set their detection limit for each subgroup assay is unclear. How genome copies equivalent was calculated?

-As the virus can be detected in other sources than feathers (for example eggs), why the authors didn´t test their assays against egg sample, for example?

#Results and Discussion:

-Did the author validate their new assays on Egg samples?

-Did the author test their assays against mixed sample (different ALV subgroups in one sample)?

-Can the authors provide few PCR curves for the results of their assays?

Minor:

-Title: “complete eradication” can be removed. It is just an opinion of this reviewer that here the authors developed a tool which can be useful for control and prevention of ALV.

-Abstract and M&M: real-time quantitative reverse transcriptase PCR or quantitative real-time PCR?

-Line 31: “two genetic subgroups” please name them.

-Line 52: Please add reference.

-Line 60: where is group F circulating? In which species?

-Line 100, 117 and the rest of the manuscript: “animals” to birds

-Line 143: can move this paragraph after line 155.

-Line 145: 50 cycles? How could the authors exclude that this large number of cycles generate false positive results?

-Line 151: the abbreviations for the primers/probe should be clarified either at the M&M or below the tables.

-Line 179: “SPF” abbreviation was not mentioned in the manuscript.

-Line 187: “We then moved on to analyze DNA from several tissues (feathers, liver, spleen,

etc.)” what is etc? please name explicitly the tissue.

-Line 28: qRT-PCR?

-Table 1: why ALV B is after ALV J?

-Table 2: as those primers/probes are already published, why the authors mentioned them in a separate table? Did the authors make any modification?

6. PLOS authors have the option to publish the peer review history of their article (what does this mean?). If published, this will include your full peer review and any attached files.

Reviewer #1: No

Reviewer #2: No

---

## [Author Response · Author response to Decision Letter 0]

1 Mar 2022

Reviewer #1: Borodin and colleagues present new PCR assays for the detection of typical exogenous Avian Leukosis Viruses present in their flocks of interest. The authors clearly present the need for PCR-based assays to detect particularly subgroups J and K, but limited data is presented as evidence for the improvements made by their tests. Tables contain redundant information, and graphical representation of reductions in overall infection rate would be most welcome. The authors need to ensure they are consistent and clear in the differences between exogenous and endogenous ALV, and that the distinction between particularly J and K needs to be made clear when discussing results. 

Of most particular concern is the repeat of an entire section of the introduction in the conclusion. This makes me doubt the integrity of the entire manuscript. 

Response: We thank the reviewer for the detailed comments on our manuscript. We address each point below.

R#1: Lines 60-2 – Make clearer the distinction between highly pathogenic exogenous viruses and the much less pathogenic endogenous group. 

Response: We modified the text accordingly, see line 14 and below (from here on, line numbers reference the new version of the manuscript).

R#1: Lines 74-5 – ALVE21/ev21 is not a gene, but an ALVE integration site, so please change this sentence. I think “ERV” will then be superfluous here, but it is the first time used (and therefore not defined). 

Response: We agree and have modified the text (line 29 and below).

R#1: Line 75 – the indicated reference 16 doesn’t talk about ALV susceptibility – is there an error here? Other works by Smith, Levin, Fadly etc from the early 1990s do talk about the importance of ALVE sequences (and not just ALVE21, but also ALVE6, ALVE9 etc) for receptor interference, but the relevance of receptor interference depends on the subgroup-specific viral entry receptors. 

Response: We thank the reviewer for catching this mistake. We substituted the correct reference.

R#1: Lines 112-113 – I don’t yet understand how you can eradicate an exogenous virus by PCR alone? Unless eradicate just means to identify infected individuals, rather than eradicating the impact from the flock entirely?

Response: We identified and removed infected individuals. We endeavored to make this clearer in the text.

R#1: Line 123,125 – GenBank, rather than GeneBank

Response: Thank you, fixed.

R#1: Line 133 – The ALV-J envelope is derived from an EAV-HP element, often numerous in the genome. What precautions were taken to ensure other, non-ALV chicken ERVs were not detected?

Response: The ALV J envelope is only partially derived from the EAV-HP element. The other part contains a portion of ev-D. Our primers flank the breakpoint, eliminating potential non-ALV J and ERV amplicons. We now illustrate this in the new Figure 1A.

R#1: Lines 170-2 – to put these types of figures on detection is interesting, but could it have context? Can you compare this directly to existing PCR and non-PCR approaches? 

OK, lines 177-179 do some comparison, but you say “in your hands” – can you give context of what has previously been reported in terms of sensitivity?

Response: We did not compare PCR and non-PCR methods for most of the experiment because the latter are unacceptably labor intensive for our large-scale applications (we added a statement to this effect on line 130). We synthesized the primers published by Qin et al. and repeated ALV detection according to their methods, but on our material (this is the meaning of "in our hands"). Our results are similar to those published by Qin et al. We added relevant clarification on line 132 and below. We did confirm PCR-negative birds using ELISA (line 179 and below) at the end of our eradication program.

R#1: Lines 181-185 – 52% seems very high! Is this what you expected? Or higher than you suspected?

Response: Given published estimates of ~20% overall flock mortality and the fact that not all infected birds die, our results seem in line with previous experience. We added discussion of this point on line 145. We were unable to find published PCR-based data on infection frequencies, however. 

R#1: Also (line 183) – switch herd for flock.

Response: Done, thank you for catching this.

R#1: Line 186 – were there any differences in detection prevalence or apparent sensitivity between tissues?

Response: We did not do any among-tissue comparisons because our main goal is a non-invasive quick detection method. Since using feathers proved to be effective, we focused on them because they are the easiest to sample.

R#1: Line 204-5 – Is the 77th generation what the authors mean? 77th round of PCR testing? No data is shown to support generational crosses?

Response: Yes, we are referring to the 77th generation from the start of the breeding program. We made it clearer in the text (line 163).

R#1: Lines 235-247 (in conclusion) are a direct repeat of the introduction, only skipping the week number in each part of the third sentence. This is a very very odd thing to do – why have the authors done this? It makes me doubt other parts of the manuscript, if copy and pasting is being used

Response: We thank the reviewer for catching this unfortunate mistake. The second instance of the paragraph has been removed.

R#1: Lines 248-9 – the authors don’t appear to show at any point any difference in detection between multiple site probes? So how can this assertion be made? It could be that one probe set detects all the time?

Response: We discuss this in the paragraph starting from line 164. We conducted separate experiments, adding the JJPLN probe and the LTR locus on top of the initial JNP probe, with increased detection at each step.

R#1: Tables1-4 – lots of the primers are repeated, creating superfluous content. Better to have one large table showing how different primers are used together and for what purpose, than many separate tables with duplicated information. A single landscape orientation page would cover it. Table legends could be more informative.

Table 5 give lots of information, but it’s really had to get the message across. A graph depicting the infection rate would be much more informative, or proportional ratios as a graph.

Response: We consolidated all the primer information into one table (new Table 1) and added a figure with infection rates (Figure 2). We did, however, retain the eradication data table, in case some readers would like to see the raw numbers.

Reviewer #2: In the manuscript entitled “Complete eradication of avian leukosis virus subgroups J and K using multilocus qRTPCR in broiler cross chickens”, the authors developed a novel qRT-PCR assay to ALV subgroups A, B, J, and K . The authors worked hardly to validate their assays but the following points should be considered and addressed by the authors in the manuscript prior to being submitted for publication as detailed below:

Response: We thank the reviewer for taking the time to provide a detailed critique of our manuscript. We respond to each point below.

R#2: -The M&M is very short and the authors should expand their methods to show for example: alignment figure where they highlight the locations of primers/probe. 

Response: We provide additional details in the M&M section. We added Figure 1, the A panel shows a schematic representation of our probe position.

R#2: -In DNA isolation and qRTPCR: how many feather samples were used? How many samples were used for RNA isolation? How many field samples? Only 10?

Response: We used 16 feathers for RNA isolation (now mentioned in M&M, line 98). All samples were field samples.

R#2: -Description for how the authors set their detection limit for each subgroup assay is unclear. How genome copies equivalent was calculated?

Response: We used cloned virus fragments from the relevant subtypes. In the case of ALV J, we also used the published test system (Qin et al., 2013) to verify our estimates.

R#2: -As the virus can be detected in other sources than feathers (for example eggs), why the authors didn´t test their assays against egg sample, for example? 

Response: Our aim was to develop a non-invasive system, so we did not use eggs. We did, however, use several tissues in our regional survey (line 145 and below).

R#2: -Did the author validate their new assays on Egg samples? 

Response: See above.

R#2: -Did the author test their assays against mixed sample (different ALV subgroups in one sample)? 

Response: We used field samples, a portion of which were double infected by ALV J and K. This is now mentioned on line 140).

R#2: -Can the authors provide few PCR curves for the results of their assays? 

Response: We added a representative set of curves in the B panel of Figure 1.

R#2: -Title: "complete eradication" can be removed. It is just an opinion of this reviewer that here the authors developed a tool which can be useful for control and prevention of ALV.

Response: Since we can no longer detect the virus in our flock, we believe "eradication" is an appropriate term. However, since our system has a finite precision, we agree with the reviewer that we cannot state that the eradication is complete. We dropped that word from the title.

R#2: -Abstract and M&M: real-time quantitative reverse transcriptase PCR or quantitative real-time PCR?

Response: It is quantitative real-time PCR. To avoid confusion, we no longer use the abbreviation in the text.

R#2: -Line 31: “two genetic subgroups” please name them.

Response: Changed the text accordingly.

R#2: -Line 52: Please add reference.

Response: Added (it is reference [1]).

R#2: -Line 60: where is group F circulating? In which species?

Response: This subgroup is specific to pheasants (we added a reference to that effect on line 17).

R#2: -Line 100, 117 and the rest of the manuscript: “animals” to birds

Response: Fixed, thank you.

R#2: -Line 143: can move this paragraph after line 155.

Response: We moved the paragraph as the reviewer suggests.

R#2: -Line 145: 50 cycles? How could the authors exclude that this large number of cycles generate false positive results?

Response: We used fresh reagents and negative controls, including specific pathogen-free chicken DNA (now mentioned on line 137), to check for false positives. We also note that false positives are less detrimental than false negatives when the goal is to eliminate infected individuals.

R#2: -Line 151: the abbreviations for the primers/probe should be clarified either at the M&M or below the tables.

Response: Primer and probe names are arbitrary and are not acronyms.

R#2: -Line 179: “SPF” abbreviation was not mentioned in the manuscript.

Response: "Specific Pathogen-Free." Now written in full (line 137).

R#2: -Line 187: “We then moved on to analyze DNA from several tissues (feathers, liver, spleen, 

etc.)” what is etc? please name explicitly the tissue. 

Response: All tissues now explicitly stated.

R#2: -Line 28: qRT-PCR?

Response: We no longer use the abbreviation.

R#2: -Table 1: why ALV B is after ALV J?

Response: Primers for all subgroups are now listed alphabetically in the table.

R#2: -Table 2: as those primers/probes are already published, why the authors mentioned them in a separate table? Did the authors make any modification?

Response: We re-synthesized the primers, using the exact published sequences. We list them here (new Table 1) so that the readers do not have to search through references for them and can verify that we in fact used the correct primers.

---

## [Decision Letter · Decision Letter 1]

29 Mar 2022

PONE-D-20-34696R1Eradication of avian leukosis virus subgroups J and K in broiler cross chickens by selection of infected birds using multilocus PCRPLOS ONE

Dear Dr. Greenberg,

Thank you for submitting your manuscript to PLOS ONE. We invite you to submit a revised version of the manuscript that addresses the points raised during the review process.

We look forward to receiving your revised manuscript.

Kind regards,

Michelle Wille

Academic Editor

PLOS ONE

Journal Requirements:

Reviewers' comments:

Reviewer's Responses to Questions

**Comments to the Author**

1. If the authors have adequately addressed your comments raised in a previous round of review and you feel that this manuscript is now acceptable for publication, you may indicate that here to bypass the “Comments to the Author” section, enter your conflict of interest statement in the “Confidential to Editor” section, and submit your "Accept" recommendation.

Reviewer #1: (No Response)

Reviewer #2: All comments have been addressed

2. Is the manuscript technically sound, and do the data support the conclusions?

Reviewer #1: Partly

Reviewer #2: Yes

3. Has the statistical analysis been performed appropriately and rigorously? 

Reviewer #1: N/A

Reviewer #2: N/A

4. Have the authors made all data underlying the findings in their manuscript fully available?

Reviewer #1: Yes

Reviewer #2: Yes

5. Is the manuscript presented in an intelligible fashion and written in standard English?

Reviewer #1: Yes

Reviewer #2: Yes

6. Review Comments to the Author

Reviewer #1: Thank you to the authors for considering my previous review suggestions. Coupled with responses to the other reviewer, I feel this manuscript has been greatly improved and would be acceptable for publication with some additional minor revisions. These comments are directed at data presentation - the figures are very welcome, but executed at a standard below the level required for this journal.

Thank you also for removing the duplicated paragraph in the conclusions, though I reiterate to the authors how disconcerting this was in the original review. Care must be taken in future to not repeat this.

1. For clarity the title should probably say "against infected birds" rather than "of infected birds"

2. Table 2 - where no birds ever co-infected with both J and K? Given your numbers (particularly at early ages) that seems unlikely? (totals in the ratio column always add up to number infected)

3. Figure 1 - a schematic is very welcome, but 1A doesn't really show anything. It would be much better to present to-scale schematics for both ALV-J and -K showing where all primers go across the elements. This would allow you to address where the conserved regions are between subgroups, and where the unique regions are that you have targeted.

4. Figure 1 - similarly, inclusion of B is good but the low resolution graph and vague supporting text means you don't take much from the graph. Which lines specifically are the negative controls? Where is your cut-off for non-specific targeting? Did you check those with lowest fluorescence for homology with desired target?

5. Figure 2 - as flock sizes differ (slightly), % infected would be more informative, with some way of indicating the difference in drop-off rate for K and J. As you are describing the flock as a whole, you could use line graphs (with separate lines for each flock, and for K/J). Anything to improve the clarity. Similarly, a 6 word figure legend is not sufficient for understanding the graph in isolation.

Reviewer #2: (No Response)

7. PLOS authors have the option to publish the peer review history of their article (what does this mean?). If published, this will include your full peer review and any attached files.

Reviewer #1: No

Reviewer #2: No

---

## [Author Response · Author response to Decision Letter 1]

13 May 2022

We thank the reviewer for the additional comments and reply point by point below.

1. For clarity the title should probably say "against infected birds" rather than "of infected birds"

 > We changed the title as suggested

2. Table 2 - where no birds ever co-infected with both J and K? Given your numbers (particularly at early ages) that seems unlikely? (totals in the ratio column always add up to number infected)

 > We did not see co-infections in this experiment. Lack of double infections is not that unlikely. For example, for the B7 Plymouth Rock line on day 140 (where the overall infection prevalence is highest), the prevalence of single infection is 0.5% (K) and 13.5% (J). Assuming independent infection, expected double-infection frequency is roughly 0.005 * 0.135 = 0.0007. This implies 0.0007*966 = 0.65 < 1 birds expected to be infected by both viruses. 

3. Figure 1 - a schematic is very welcome, but 1A doesn't really show anything. It would be much better to present to-scale schematics for both ALV-J and -K showing where all primers go across the elements. This would allow you to address where the conserved regions are between subgroups, and where the unique regions are that you have targeted.

4. Figure 1 - similarly, inclusion of B is good but the low resolution graph and vague supporting text means you don't take much from the graph. Which lines specifically are the negative controls? Where is your cut-off for non-specific targeting? Did you check those with lowest fluorescence for homology with desired target?

 > See the new Figure 1. We believe we addressed all these suggestions.

5. Figure 2 - as flock sizes differ (slightly), % infected would be more informative, with some way of indicating the difference in drop-off rate for K and J. As you are describing the flock as a whole, you could use line graphs (with separate lines for each flock, and for K/J). Anything to improve the clarity. Similarly, a 6 word figure legend is not sufficient for understanding the graph in isolation.

 > See the new Figure 2. We implemented the changes the reviewer suggests.

---

## [Editor Report · Decision Letter 2]

24 May 2022

Eradication of avian leukosis virus subgroups J and K in broiler cross chickens by selection against infected birds using multilocus PCR

PONE-D-20-34696R2

Dear Dr. Greenberg,

We’re pleased to inform you that your manuscript has been judged scientifically suitable for publication and will be formally accepted for publication once it meets all outstanding technical requirements.

Kind regards,

Michelle Wille

Academic Editor

PLOS ONE
---

## [Editor Report · Acceptance letter]

16 Jun 2022

PONE-D-20-34696R2 

Eradication of avian leukosis virus subgroups J and K in broiler cross chickens by selection against infected birds using multilocus PCR 

Dear Dr. Greenberg:

I'm pleased to inform you that your manuscript has been deemed suitable for publication in PLOS ONE. Congratulations! Your manuscript is now with our production department. 

Kind regards, 

on behalf of

Dr. Michelle Wille 

Academic Editor

PLOS ONE